# Innovative Smart Drilling with Critical Event Detection and Material Classification

**Kantawatchr Chaiprabha and Ratchatin Chancharoen ***

Department of Mechanical Engineering, Faculty of Engineering, Chulalongkorn University,
Bangkok 10330, Thailand; 6570016221@student.chula.ac.th
* Correspondence: ratchatin.c@chula.ac.th; Tel.: +66-82-218-6643

**Abstract:** This work presents a cyber-physical drilling machine that incorporates technologies discovered in the fourth industrial revolution. The machine is designed to realize its state by detecting whether it hits or breaks through the workpiece, without the need for additional sensors apart from the position sensor. Such self-recognition enables the machine to adapt and shift the controllers that handle position, velocity, and force, based on the workpiece and the drilling environment. In the experiment, the machine can detect and switch controls that follow the drilling events (HIT and BREAKHTROUGH) within 0.1 and 0.5 s, respectively. The machine's high visibility design is beneficial for classification of the workpiece material. By using a support-vector-machine (SVM) on thrust force and feed rate, the authors are seen to achieve 92.86% accuracy for classification of material, such as medium-density fiberboard (MDF), acrylic, and glass.

**Keywords:** drilling; actuation; drilling breakthrough; artificial-intelligence algorithm; drilling control

## 1. Introduction

Mechanical drilling machines, which were developed during the second industrial revolution, allowed for greater power, speed, and precision in drilling [1]. Such drilling machines have been used in a wide range of industries, including petroleum, wood, automotive and construction [2]. The computer numerical control (CNC) milling machines, developed during the third industrial revolution, are considered advanced drilling machines [1,3]. By combining existing drilling technology with robotics and actuation, these machines are computer-controlled and can carry out complex operations [4]. During the process of cutting, lubrication is used to reduce friction and heat generated, enabling a cut to be made along a programmed trajectory [4,5]. Thereby, the efficiency, accuracy, precision and the tool's lifespan are significantly improved [6]. In modern industries today, these CNC machines play a vital role in the aerospace industry, medical devices, electronics, research development and consumer goods [6–12]. In this paper, the drilling machines used are further advanced with AI, IoT, big data, and cloud computing, and have a significant input in smart factories [13].

Cyber-physical systems (CPS) are systems capable of making decisions and can operate independently. Such CPS systems combine sensor networks with embedded computing to monitor and control the physical environment. These systems involve feedback loops that can self-activate control and are seamlessly integrated towards performing well-defined tasks. CPS is a core of industry 4.0 that emphasizes the interaction between digital and physical processes [14–17]. As such, industry 4.0 refers to smart systems that are connected to production systems designed to sense, predict, and interact with the physical world to make decisions that support production in real-time. Industry 4.0 can increase production, energy efficiency, and sustainability, but require vertical integration in various fields: mechanical engineering, electronics, control systems, and computer science. The seamless connection in CPS allows for collecting data from the real environment, then, suitable decisions are selected and enacted in real-time [16,18,19]. CPS features advanced

actuation and sensing [1], in-process measurement [20], real-time health monitoring [21], life assessment [1,11,20], self-adaptation [22], and collaboration with artificial intelligence (AI) [1,23,24]. The aim of this project is to develop a novel drilling machine that incorporates technologies discovered in the fourth industrial revolution such that it can revolutionize the drilling process.

A drilling machine consists of a drill bit, a spindle actuator, and a feed actuator [2]. During operation, the spindle rotates the drill bit at a fast constant speed while the feed actuator provides a thrust force against the workpiece [2]. It is noted that the knife of the drill bit generates mechanical shear force that exceeds the strength of the workpiece and cuts off chips [25]. The chips are then removed through spiral flutes on the drill bit [26]. The process of drilling is complex and thus widely studied and modeled to achieve the desired hole quality [25,26]. Factors that can affect the quality of a hole include cutting speed, feed rate, cutting force, cutting fluid, and drill bit geometry [6,8,27–29]. Experts typically choose the best options and parameters to optimize the process [2,30]. Drilling can create low residual stresses around the hole opening and high stresses on the hole's cylindrical surface. As drill bits are in continuous use, the surface becomes degraded as its surface always bears friction. This progressive wear affects the quality of drilling and is one of the challenges that is constantly being researched [11].

In the proposed cyber-physical drilling machine, the total machine is re-engineered to effectively utilize the newly discovered technologies, as mentioned previously. In a typical drilling machine, both the rotational motor and ball screw are used to drive the spindle carriage. Herein, the rotational motor and ball screw are replaced with a powerful direct drive brushless linear motor. In this way, the current command is highly related to the thrust force and the spindle carriage is fully backdrivable. Thus, mechanical advantage is traded off with greater sensibility and backdrivability. Drawbacks are suppressed by advanced power electronics and a motor design using a rare-earth magnet. Actuation and sensing are now in the same mechanism.

In the proposed conceptual design, machine intelligence can directly sense both the feed force and the motion of the spindle carriage in process. Abnormalities and faults are also monitored. One feature that stands out is the real-time hit and breakthrough detection that interrupts the embedded control during the process of drilling. Initial contact between the drill and workpiece (hit) can cause delamination in carbon-fiber-reinforced-polymer composites [12,29,31]. Several studies have investigated techniques to detect and handle breakthroughs, especially in the orthopedic field where vital research has been carried out to predict bone drilling's breakthrough to prevent tissue damage [32–37].

The main contribution of this paper is a novel drilling machine that combines new perception, intelligence, control, and actuation techniques to achieve superior flexibility, adaptability, efficiency, and safety. For perception, such a machine can sense not only the thrust force and position but also the workpiece's mechanical properties, process state, hit, and breakthrough. For intelligence, AI is used to identify both the workpiece's material and process state and can handle the process's parameters for a known material. For control, either force or position can be applied. For actuation, the feed drive is a direct-drive linear brushless motor with advanced current drive. The absence of mechanical transmission means that there is no nonlinear asymmetric behavior. As such, this new and authentic work investigates the performance of the proposed drilling machine. Our current findings expand upon scant previous works. This cutting-edge prototype provides new insights into previously unexplored areas.

## 2. Analysis of the Transmission in Typical Actuation

A servo motor is an actuator, which produces rotational motion and provides mechanical torque. A servo motor allows for precise control and can move parts of a machine with great efficiency. A lead screw is used to convert rotation into translation. The screw provides mechanical advantage for both the drive force and the resolution of the position sensing. The screw is typically used with a fitted linear bearing.

### 2.1. The Mechanism of a Lead Screw

In Figure 1, a model of a lead screw, commonly used in a drilling machine, is illustrated. The mechanism consists of a screw and a carriage, which are connected through spiral teeth. Herein, the mechanism, in cylindrical coordinate, is transformed into a planar problem. In this transform, the rotational motion of the screw θ rad is represented by the linear motion θr m, where r is the radius of the screw in meters. If x is a position of the carriage in meters and $\alpha$ is a lead angle of the screw in radians, then the relationship can be expressed as:

$$x = \tan(\alpha) \cdot \theta r \tag{1}$$

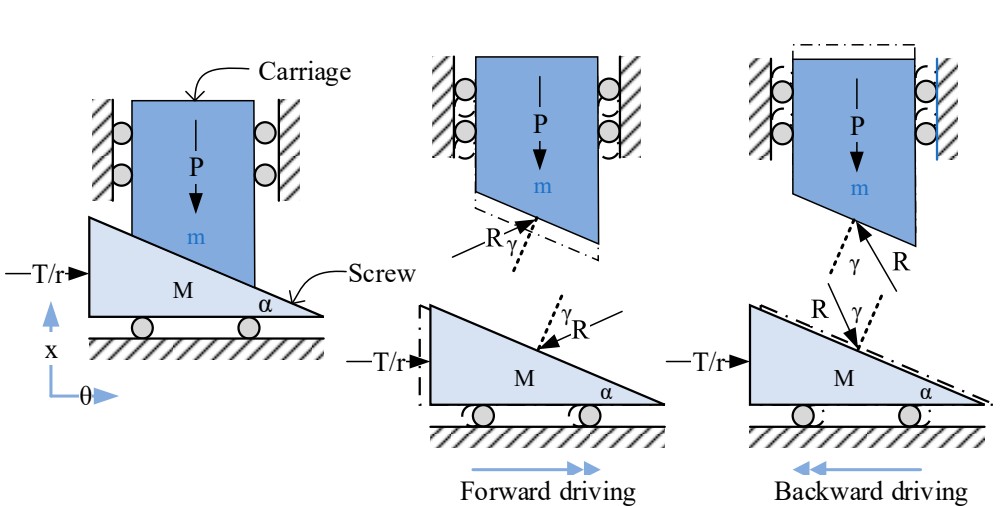

**Figure 1.** Force analysis of the wedge model of a lead screw with lead angle ($\alpha$) and radius (r). The blue shape is a carriage, the light-blue shape is an imaginary unwrapped spiral tooth of screw, and x and θ are translational and rotational motion axis, respectively. The mass of screw and carriage are M and m, respectively. Actuation torque is denoted by T and load force is denoted by P. The resultant reaction force between screw and carriage is denoted by R. Two directions of friction cone γ are shown, depending on the direction of motion.

Friction between the contact surfaces of the lead screw is represented by the friction cone (γ). Hence, γ can vary based on the driving direction of the mechanism as shown:

$$\gamma = \begin{cases} -\arctan(\mu_k), & \text{backward drive} \\ \arctan(\mu_s), & \dot{y} = 0 \\ \arctan(\mu_k), & \text{forward drive} \end{cases} \tag{2}$$

where $\mu_s$ and $\mu_k$ denote the static and kinetic friction coefficients, respectively.

The equation of motion can be defined as:

$$\frac{T}{r} - R\sin(\alpha + \gamma) = M\,r\,\ddot{\theta} \tag{3}$$

$$R\cos(\alpha + \gamma) - P = m\,\ddot{x} \tag{4}$$

where M and m are inertia of a screw and nut, respectively and R is the resultant force of normal force and friction at a surface.

Combining Equations (1), (3), and (4), the equation of motion yields:

$$\left(M\,r^2 + m\,r^2\tan(\alpha + \gamma)\tan(\alpha)\right)\ddot{\theta} = T - P\,r\tan(\alpha + \gamma) \tag{5}$$

In statics, the mechanical advantage (σ) is equal to:

$$\sigma = \frac{1}{r}\cot(\alpha + \gamma), \tag{6}$$

$$P = \sigma T \tag{7}$$

Herein, backward driving gives rise to σ because γ is negative. Conversely, during forward driving, σ decreases. It is noted that if the sum of α and γ is less than zero, the screw cannot backdrive and exhibits self-locking.

The direction of actuation is significant. Two influential factors are found; namely, the direction of motion, and the surface where the screw makes contact. As observed in Figure 1, the direction of motion can alter the direction of the friction cone (γ). Additionally, since a lead screw has two spiral contact surfaces, switching the contact surface results in a switch of the actuation direction. Such an effect can be expressed as:

$$\gamma = \begin{cases} \arctan(\mu_s), & \dot{y} = 0 \\ sgn\left(N{\cdot}\dot{\theta}\right)\arctan(\mu_k), & \dot{y} \neq 0 \end{cases} \tag{8}$$

where N is a normal force and can be written as:

$$N = R\cos(\gamma) \tag{9}$$

where N represents the surface that is in contact. By combining Equations (1), (4), (5), and (7), N can be derived, as follows:

$$N = \frac{\cos(\gamma)}{\cos(\alpha + \gamma)}\left[\frac{\left(MP + \frac{T}{r}m\tan(\alpha)\right)}{M + m\tan(\alpha + \gamma)\tan(\alpha)}\right] \tag{10}$$

It is evident that N depends on both actuation (T) and load (P). As such, load (P) can disturb the direction of actuation and mechanical advantage.

### 2.2. The Visibility of a Lead Screw

Visibility refers to the degree of clarity whereby the input side of a mechanism can observe what happens on the output side, such that it can be categorized as clear, obscured, or opaque. The focus of this study is on the visibility of the load force that can be detected on the drive side. An actuator can perceive the effect of the load force, enabling the machine to receive vital information. Thus, the machine is able to ensure proper operation.

A lead screw is used in machines for high-precision actuation. A screw mechanism, which is a type of mechanical power transmission, enlarges force output by multiplying its mechanical advantage with input. Such an effect increases the resolution of force output. A resolution is the smallest discrete change in output. By expanding this, it increases ambiguity. This effect is considered a downside for visibility. As a result, the resolution of the force worsens, resulting in poor visibility.

The mechanical advantage of a lead screw can amplify the effect of Coulomb friction by expanding the friction dead-band on the driven side. This outcome occurs because it intensifies the Coulomb friction that is produced from the driving side. Consequently, it is more difficult for the load to exceed overall Coulomb friction. It is significant that the load value cannot be measured within the dead-band due to the lack of any response. Besides, a decrease in backdrivability aggravates the friction dead-band by widening the dead-band gap. In the case of a self-locking lead screw, the effect of the load is reduced to zero, resulting in an infinitely large friction dead-band, which blocks all visibility. When the screw is still moving, there is no effect of discontinuity from Coulomb friction. However, such motion can exhibit non-linearity caused by the direction of actuation (Figure 2).

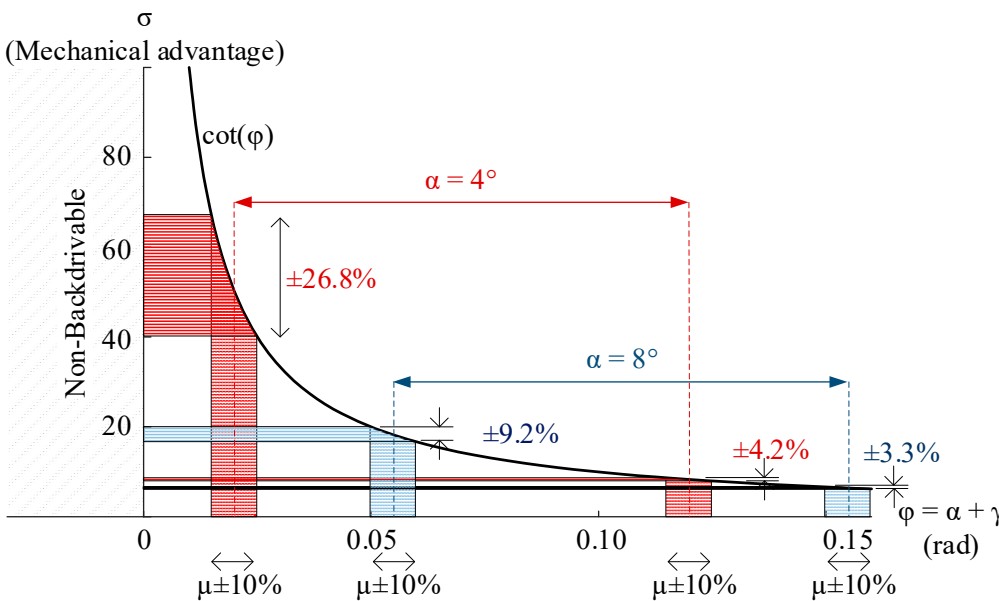

**Figure 2.** The dynamic visibility and mechanical advantage of a lead screw.

In Figure 2, mechanical advantage is affected by the perturbation of friction. Such fluctuation implies the inaccuracy of the force output as it directly depends on mechanical advantage. In Equation (6), mechanical advantage is defined as the cotangent of summation between the lead screw angle and friction angle ($\alpha + \gamma$). Therefore, the fluctuation regarding ($\alpha + \gamma$) can grow larger as it gets closer to zero. This effect also depends on the direction of actuation. To simplify the analysis, the chosen lead screw is assumed to have a radius of 1 m. With a 4° lead angle and 0.1 friction coefficient, a disturbance of 10% can produce an error of 26.8% for mechanical advantage in backward driving, thus increasing the disturbance to 268%. In contrast, forward driving can reduce the error to 4.2%. As a result, perturbation decreases. When a higher lead angle (8°) lead screw is used, the perturbation effect is reduced to 9.2% and 3.3% for backward and forward driving, respectively. It is evident, therefore, that the effect of perturbation can vary, depending on driving direction and mechanical advantage. In backward driving, a high mechanical advantage lead screw, which has a low lead angle, tends to produce low visibility as it greatly augments the effect of friction perturbation.

In summary, this discussion has highlighted the various effects of lead screws on visibility. Factors such as resolution, Coulomb friction dead-band, variation of mechanical advantage, and friction perturbation, can all contribute to the loss of visibility.

### 2.3. Effect of Lead Screw on a Feedback Controller

In Figure 3, a closed-loop control diagram for the feed of a drilling machine is shown. The controller commands the current drive, which provides the power to the motor. Hence, the drilling machine has a permanent magnet motor such that the resulting torque is highly related to the drive current [38]. A lead screw provides mechanical advantage and converts the rotational motion of the motor into translation of the spindle's carriage [39–41]. Mechanical advantage amplifies the driving force but reduces the effect of driving motion [42,43]. Such an effect is seen to enhance the resolution of the motion sensor on the input side. Moreover, the screw mechanism significantly decreases the degree of back-drivability. Thus, the load is less disturbed on the drive side. The encoder is installed on the drive side and provides feedback to the closed-loop position controller. It is noted that the screw causes significant friction. Subsequently, when the carriage moves, the kinetic energy is dissipated, resulting in greater control and stability. Overall, the architecture of this closed loop is excellent in terms of closed-loop position control, and is widely used in current designs.

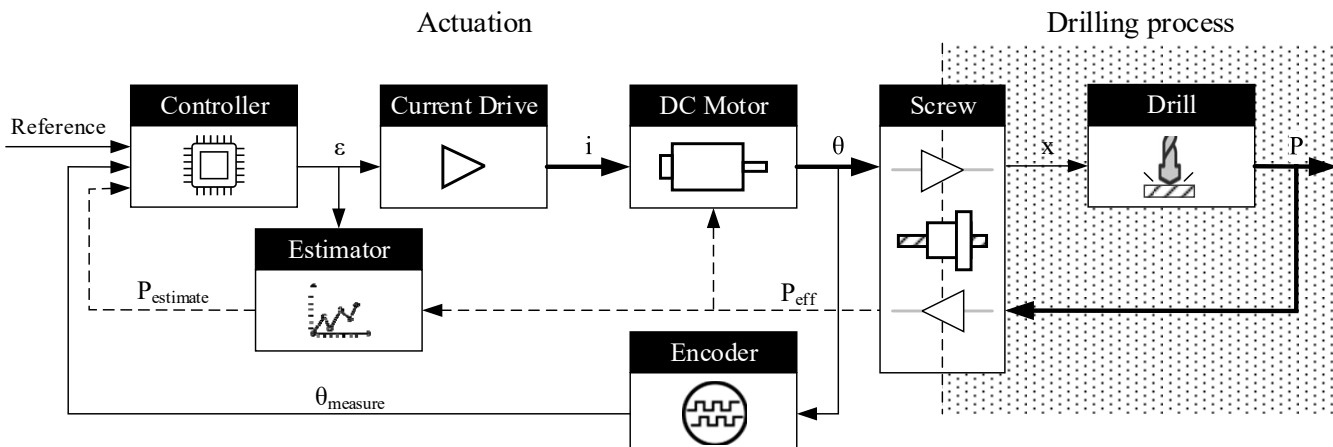

**Figure 3.** Schema of control diagram: conventional feeding axis of a drilling machine.

### 3. Design and Implementation

#### 3.1. Conceptual Design

In our proposed architecture, a sense of feed force is desired during a drill's operation. Although a screw offers several benefits, it blocks the sense of the feed force. Visibility during actuation is also of concern in our design. The behavior of the screw is asymmetric, and visibility depends on the direction of the actuation. When the load forces the system in a backward direction, visibility worsens. During actuation, HIT and BREAKTHROUGH events are monitored and feed force is sensed. Moreover, the screw is expelled during actuation to gain full visibility.

In our proposed design, several control drawbacks can be seen: higher disturbance, lower stability, and less actuated force. However, both disturbance and instability are mitigated with advanced computing power that delivers more complex mathematics at a high-frequency update rate. The low actuated force caused by the removal of transmission is compensated for by having a rare-earth magnet motor and a modern power electronic circuit. Such a magnet provides better torque constant and better current flow. In this way, modern technology can compensate for all drawbacks of the design.

#### 3.2. Hardware Implementation

As shown in Figure 4, the drilling machine has a linear brushless DC motor (RCL-SA3L). The motor is set up vertically as a direct-drive motor. At the front of the linear motor, a spindle motor (24 V, 21,000 rpm) is located. Below the spindle, both the drilling chuck and bit are fixed. Thus, the motion of the drill bit along the feed axis is in translation with the carriage of the feed motor. Its position is sensed by a linear incremental encoder, which is embedded inside the RCL-SA3L linear motor. The resolution of the system is 0.042 mm and has a range of 64 mm. The brushless motor driver and advanced motion controls (AMC) has a 24 V power supply. The spindle assembly can be driven in the feed direction with a programmable torque while its position is sensed in real time. The motion of the spindle assembly is real-time controlled by a 32-bit microcontroller. Such a microcontroller can sense the encoder's signal, generating simulated hall signals, and can handle a six-phase commutation of the brushless motor with interrupt-driven programming. The actual hardware can be observed in Figure 5.

#### 3.3. Proposed Controller

In Figure 6, our AI-embedded drilling machine consists of three states: IDLE, FEED and DRILLING. The machine starts in IDLE state. If DRILLING is commanded, then the machine switches to FEED state wherein velocity control is implemented. Next, the spindle assembly moves towards the workpiece at a constant velocity until HIT occurs. When the HIT event is implemented, it must be rapidly detected in real-time: a suitable control

strategy needs to be activated to suit the event. Thus, the operator needs to use his/her eyes to monitor the motion of the drill and must proceed with caution. The moment before the drill bit contacts the workpiece is the most critical. This moment is called HIT. When the tip of the drill bit makes contact, the force makes an impact collision. At this point, the process is unmanageable. There is the possibility that the drill bit is broken, the center of the drill bit misaligned, or a crack may appear on the workpiece.

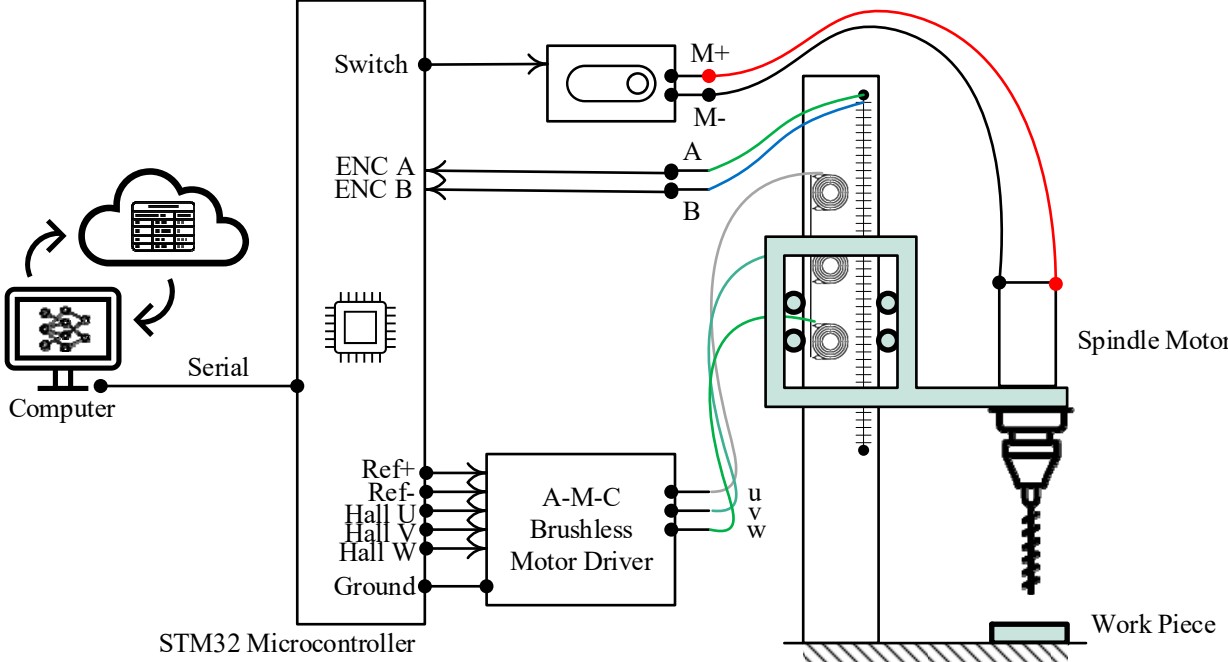

**Figure 4.** The architecture of the novel drilling machine.

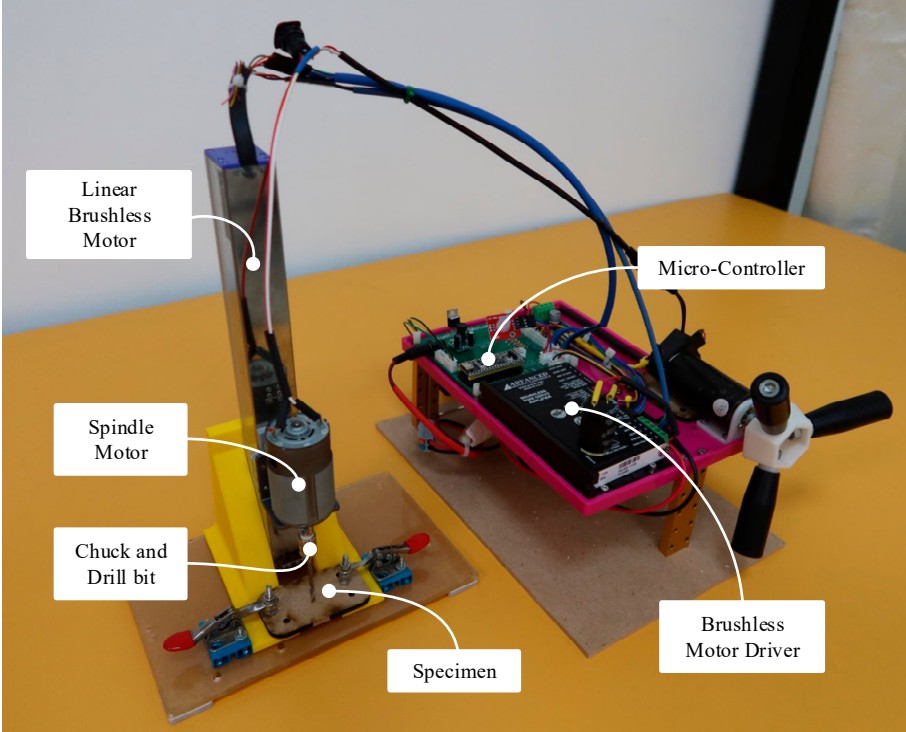

**Figure 5.** The actual implementation of a cyber-physical drilling machine.

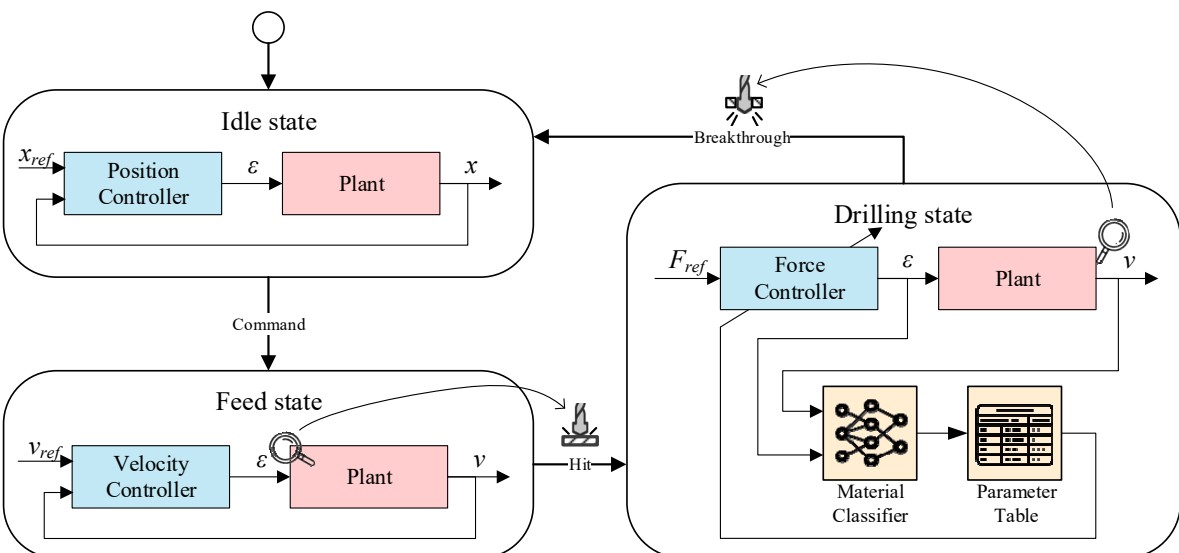

**Figure 6.** The proposed controller for the AI-embedded drilling machine. Blue blocks are processed real-time on micro-controller (STM32F411). Yellow blocks are processed via an edge computer and cloud system. The plant (red) represents a linear motor and a driver, which drives the drill.

During the cutting process, the controller switches from the HIT event to the state of DRILLING. Consequently, where the material is rigid, brittle, and fragile, force control is recommended to limit the force applied to the workpiece to avoid defects. Hence, the feed rate is not uniform. In contrast, if velocity control is implemented instead, the reaction force fluctuates and may exceed the strength of the material: this initiates cracks appearing [29]. In the proposed drilling machine, AI is used to classify the material. Then, the system will apply suitable parameters to control the drilling. Drilling is continued until BREAKTHROUGH is detected.

BREAKTHROUGH is the moment when the drill bits fully penetrate through the workpiece. At this moment, the force is not in static balance. The drill bit may accelerate with potential energy and higher speed. The challenge is that the BREAKTHROUGH should be rapidly detected in real-time, and a suitable control strategy activated to suit the event. In the experimental case, upon BREAKTHROUGH, the controller switches the state into IDLE and the spindle assembly is controlled to go to the HOME position. Herein, a simple algorithm to detect HIT and BREAKTHROUGH is proposed.

In Figure 6, controllers (blue blocks) in all states are processed real-time at 1000 Hz on microcontroller (STM32F411). The microcontroller can communicate with the motor driver to drive the linear motor and thereby monitors the incremental encoder to measure position. Control signals (position, velocity, and force) are sent at 1000 Hz to the edge computer via serial communications with 115,200 baud rates.

For our prototype, material properties such as machinability needs to be considered. Machinability is about the properties of the material, i.e., how difficult it is to cut or drill. Several factors are involved; namely, cutting force, tool wear, chip formation and disposal, work hardening, and thermal conductivity. To model such complex behavior, artificial intelligence is used. To classify material such as glass, MDF or acrylic, the SVM algorithm is also applied. For each material during drilling, the labelled data pair of feed force and velocity is employed as a training dataset. The data implemented has been preprocessed by excluding data that was not at a steady feed rate.

During the events that occur in drilling, the controller must switch behavior. Thus, the situation becomes complex. In the process of modularization and state transition, a finite state machine approach is seen to be favorable. Therefore, the definition of each state behavior and transition must be clearly defined being more readable and manageable. To handle all real-time signals and to monitor events, multi-task and multi-rate concepts

are preferred. Subsequently, control signals from the micro-controller are relayed to the edge-computer via serial communication and sent to the cloud via MQTT. Such data is stored in cloud-based big data and this is where the SVM algorithm for material prediction carries out all operations.

## 4. Results and Discussion

### 4.1. Experimental Setup

In the experiment, the proposed drilling machine was employed to drill through 10 specific specimens of acrylic, MDF, and glass (Table 1). Each specimen was securely locked on the base of the drilling machine. For acrylic and MDF, a 3 mm diameter high-speed steel drill was used. For glass, a 3 mm diameter glass drill bit was used. During the experiment, the spindle motor was constantly supplied with 24 V DC. Prior to the experiment, both HIT and BREAKTHROUGH threshold values were determined (Figure 7).

**Table 1.** Nominal thickness for each material.

| Material | Nominal Thickness (mm) |
|---|---|
| MDF [1] | 4.0 |
| Acrylic [2] | 4.0 |
| Glass | 1.2 |

[1] MDF's density is 800 kg/m$^3$. [2] Poly (methyl methacrylate) (PMMA).

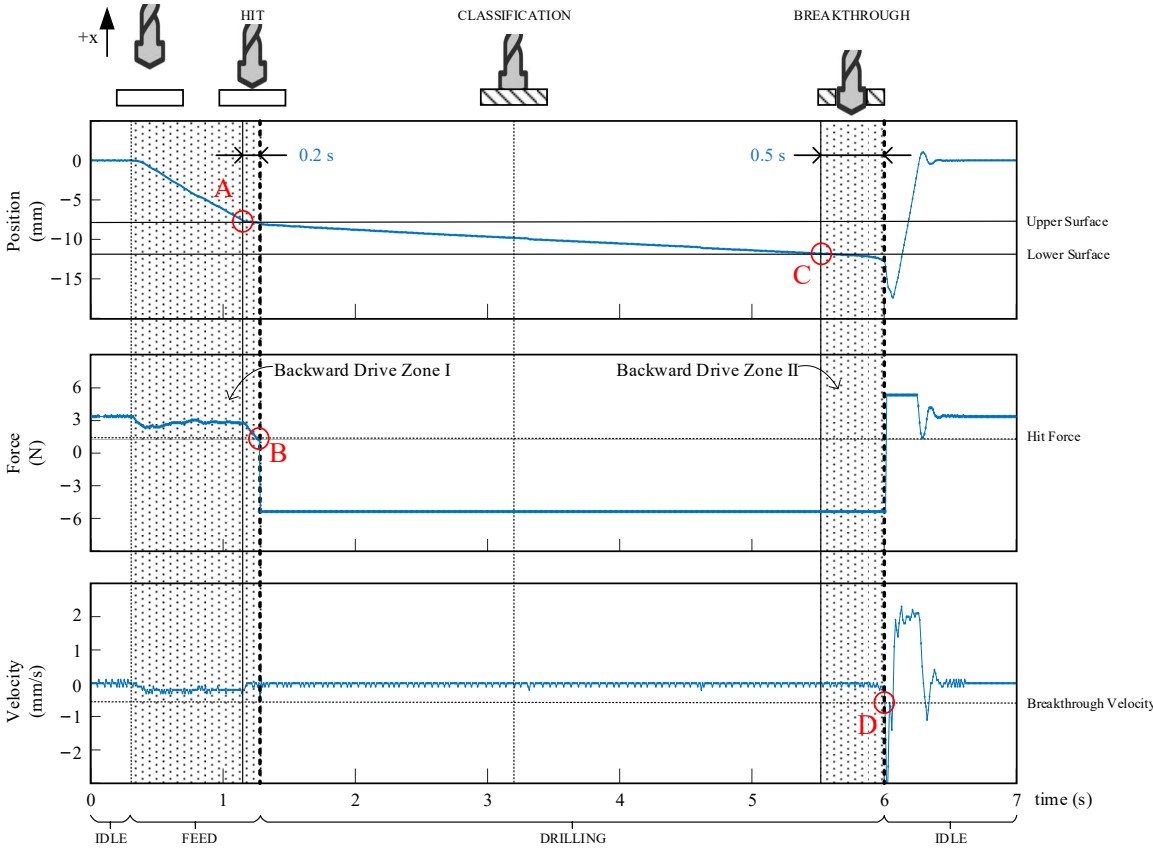

**Figure 7.** Schema of the experimental behavior of the cyber-physical drilling machine on drilling MDF. Position A refers to the initial contact between the drill and workpiece. Position B denotes the time when HIT is detected. Position C signifies when the tip of the drill bit emerges from the workpiece. Position D refers to when the drill's velocity exceeds the breakthrough threshold, thereby BREAKTHROUGH is detected.

*4.2. HIT and BREAKTHROUGH Detection*

In Figure 7, the drilling process begins with Home positioning, where the spindle is parked under stiff control. The stiff control is able to maintain the spindle at the Home position. To balance the gravity of the spindle's assembly, the drive force is kept in the vertical position. Due to the high visibility of the drive mechanism, vibration from the spindle can disturb the position controller, leading to slight oscillations in the spindle's velocity. However, the stiff PD control with gravity compensation can effectively control the position of the spindle assembly.

When the drill approaches the workpiece, velocity control mode is activated, using stiff P with gravity compensation while HIT is monitored. Then, the spindle moves towards the workpiece at a constant speed. Hence, the net required force is seen to reduce but continues oscillating; however, downward velocity is maintained. When the drill bit hits the workpiece (A), its position shifts, producing a steep slope decrease in velocity. As the reaction force of the workpiece counteracts the weight, the required force decreased. In Figure 7, the HIT threshold is denoted by thrust force (1.32 N). At Position B (1.2 s), the thrust force is monitored across the threshold. At this point, HIT is detected, and the machine is seen to switch to the drilling state. In this experiment, the threshold method proved its effectiveness, successfully detecting the HIT event with a delay of only 0.1 s.

After HIT occurred, it is noted that the thrust force suddenly changed to a constant value at 5.3 N. Thus, to monitor the BREAKTHROUGH, the machine switched to force control mode; the drill's velocity remained constant. Once the drill penetrated through the lower surface (C), the spindle accelerated due to the absence of the reaction force. At Position D (6.01 s), velocity rapidly increased and exceeded the BREAKTHROUGH threshold ($-0.5$ mm/s). At this point, BREAKTHROUGH was detected. It is evident that the velocity threshold technique was sufficient to monitor the BREAKTHROUGH. BREAKTHROUGH was detected about 0.5 s after the drill passed through the lower surface. Hence, BREAKTHROUGH was successfully monitored.

After completion of BREAKTHROUGH, the machine reverted to IDLE state with stiff PD and gravity compensation. The characteristic response recorded an overshoot of 0.96 mm and a settling time of 0.4 s. Besides, the force was saturated at 5.3 N. Velocity became saturated at around 92.6 mm/s. As settling time elapsed, the system returned to its steady state: the same level as at the beginning.

Subsequently, it was found that the Backward Driving Zone could be estimated by the direction of force and motion. In Backward Driving Zone I, the direction proved to be opposite to the thrust force. In Zone II, the drill bit gradually broke through the lower surface; the upward reaction from the material gradually disappeared. Consequently, only the weight of the drill (load force) is pushed down. Such an outcome may lead to the possibility of entering backward drive direction as the carriage moves under the effort of load. Backward Driving Zone I incorporated the feed state and the HIT event. Backward Driving Zone II included the BREAKTHROUGH event. In this situation, if the machine was installed with a lead screw, it could result in poor visibility.

In drilling operations, the proposed drilling machine with HIT and BREAKTHROUGH detection may also be used with engineering materials such as metal and ceramics. Both detection techniques monitor the change of the reaction force. In Figure 7, the experiment was conducted using material that is simple to drill (MDF). It is known that the thrust force (6 N) is relatively low compared with other drilling processes used in commercial enterprises [12,44]. In fact, general engineering materials need to have high strength; thus, more thrust force is required [12]. Such a demand intensifies force differences, thereby facilitating both detection techniques. However, the application of high thrust force may cause adverse effects since it produces high acceleration when the drill breaks through. This situation can be counteracted by increasing sampling rate and position sensor resolution.

### 4.3. Material Classification

Herein, the dataset was collected by the proposed drilling machine applying a sample rate of 1000 Hz. The dataset was selected within the drilling state between the HIT and BREAKTHROUGH. After that, an average window with 0.025 ms was carried out on both the thrust force and velocity. In total, there were 351 data points (MDF: 51, acrylic: 102, and glass: 198) as shown in Table 2. Finally, the dataset was split into a train dataset and a test dataset having a ratio of 80:20.

**Table 2.** The number of datapoints used during the experiment.

|         | Train | Test | Total |
|---------|-------|------|-------|
| MDF     | 39    | 12   | 51    |
| Acrylic | 79    | 23   | 102   |
| Glass   | 163   | 35   | 198   |

In Figure 8a, the correlation between the thrust force and velocity is depicted. As such, the correlation can be used to classify the material among the different specimens. For this study, three materials were used: namely, MDF, acrylic, and glass. As observed, dots indicate each data point. For instance, red, green, and blue indicate the ground truth for MDF, acrylic, and glass, respectively. Moreover, the red, green, and blue contours highlight SVM's predicted result and decision boundaries.

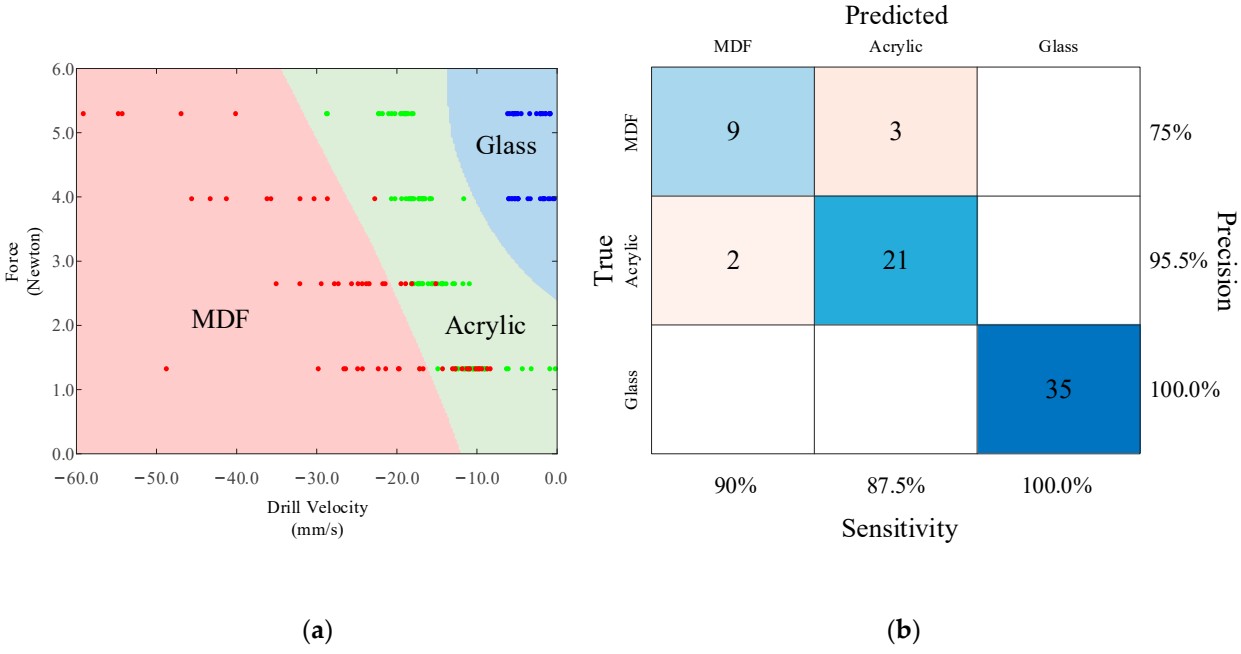

(**a**)  (**b**)

**Figure 8.** Schema of material classification using drilling velocity and force: (**a**) Scatter plot of all data points and decision areas of SVM classifier (red: MDF, green: acrylic, and blue: glass), and (**b**) Confusion matrix from test dataset.

In Figure 8a, plots show a clustering pattern that can be used to classify the material. It is noted that hard materials require high force to achieve high velocity. Glass, which has low machinability, tends to fall in the category of low velocity. On the other hand, MDF can be easily processed, which results in the scatter plot being skewed towards the left-hand side.

This dataset was then used to train a SVM model with a quadratic kernel function. The decision boundary of the model is shown in the graph as a contour color. When predicting the test dataset, the model achieved an accuracy of 92.86%.

In Figure 8b, there are false classifications regarding acrylic and MDF. When thrust force and drill velocity is low, the data points of acrylic and MDF cannot be separated (Figure 8a). When drilling with low thrust force, drill velocity is consequently slow. Velocity data, in this work, was calculated from the derivative of position as measured via an incremental encoder. As a result, noise due to vibration of the drill is greatly amplified. Furthermore, the encoder's resolution is seen to provide more influence since the resolution in comparison with true value is more significant.

Furthermore, estimation from classification can be used for developing digital twins [45]. The properties of the digital twin's workpiece can be identified via estimation results. As such, the behavior of the drilling process can be predicted and compared, thus enabling anomaly detection, health monitoring, and predictive maintenance [16].

## 5. Conclusions

In this paper, results demonstrate that the drilling machine developed can successfully detect both HIT and BREAKTHROUGH during drilling. To suit the drilling process, the machine can operate independently, thereby force or position controllers can easily be selected. Our machine can detect HIT and BREAKTHROUGH events within 0.1 and 0.5 s, respectively. According to force and motion data during drilling, AI could classify materials into MDF, acrylic, and glass. In the experiment, real-time classification using SVM attained an accuracy of 92.86%. Overall, the proposed design shows promise for use in hybrid manufacturing, where HIT and BREAKTHROUGH are essential. Real-time classification of material can benefit the drilling of laminated composite material, as it can perform measurement-based adaptive drilling. This work breaks new ground for the development of high performing event-based manufacturing machines.

**Author Contributions:** Conceptualization, K.C. and R.C.; methodology, K.C.; software, K.C.; validation, K.C.; formal analysis, R.C.; investigation, R.C.; resources, K.C. and R.C.; data curation, K.C.; writing—original draft preparation, K.C. and R.C.; writing—review and editing, K.C. and R.C.; visualization, K.C. and R.C.; supervision, R.C.; project administration, R.C.; funding acquisition, R.C. All authors have read and agreed to the published version of the manuscript.

**Funding:** This project is funded by National Research Council of Thailand (NRCT).

**Acknowledgments:** Authors would like to thank Natthapong Angsupasirikul for useful recommendations.

**Conflicts of Interest:** The authors declare no conflict of interest.

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
