# Peer review of "Innovative Smart Drilling with Critical Event Detection and Material Classification"

_jmmp, doi:10.3390/jmmp7050155_

Round 1

Reviewer 1 Report

Very interesting approach and result. Some minor corrections are needed. All my suggestions can be found in the attached PDF.

Author Response

Thank you for your valuable comment. Your concerns have been addressed. The manuscript with the markup change is in the attachment.

Reviewer 2 Report

The manuscript " Towards Industry 4.0: event-based drilling machine coordinated with AI for material classification” presents a cyber-physical drilling machine designed to detect whether it hits or breaks through the workpiece, without the need for additional sensors apart from the position sensor.  The findings were compared, and conclusions were made.

The manuscript is original and has merit, but it needs to be improved.

The Title does not present the paper’s content.

The abstract does not clearly state the authors' methodology, results, and conclusions.

The introduction does not bring in chronological order the development of the technologies used for the machining of materials and the existing academic contribution to the development of this work. The introduction brings the objective of the research and explains, and the design proposal of a machining machine without the use of auxiliary sensors. This should be described in the methodology and not in the introduction. The introduction should provide a good review of the literature in the area in which the work is inserted and what would be the major contributions of the work to the academic community.

The methodology needs to clearly outline how the prototype equipment was developed.

The parameters used for testing need to be clearly described.

The figure's caption needs to explain the Figure's content.

Moderate editing of English language required

Author Response

Thank you for your valuable comment. Your concerns have been addressed. Please see the attachment.

Reviewer 3 Report

In the article under review, the authors present the results of a study aimed at improving the functioning of the drilling machine. The main idea of this paper is the implementation of switching a controlled coordinate (position, velocity or force) depending on the workpiece and the drilling environment.

The authors presented a description of the transmission of the drilling machine and the traditional control system of its electric drive. As a result of the performed analysis, a concept for the implementation of a modernized drilling machine was proposed. The results of experimental studies are presented and their analysis is given.

However, during the review, I drew attention to the following shortcomings, the correction of which would improve the quality of the paper:

1.      In the Introduction section, there is no critical analysis of works on the research topic. In this section, the authors only formulated the purpose of the paper. The section needs to be substantially revised.

2.      The main innovation in the paper is the use of a neural network to obtain a “digital twin” of the drilling environment. However, the paper does not describe this in sufficient detail. The paper needs to be completed.

3.      In the title of the article, the authors use the terms “industry 4.0” and “AI" (artificial intelligence). However, these facts are not sufficiently reflected in the content of the paper. One gets the feeling that the authors used them only to decorate the title of the work.

In general, in the presented form, the paper has a descriptive character. It seems to me that this is not enough for publication in a high-ranking scientific journal. The authors should more clearly formulate and present the scientific novelty of the presented developments, and detail their presentation. Thus, the paper can be recommended for publication only after major revision. 

Author Response

(The authors gave the same response as above.)

Reviewer 4 Report

The paper presents some interesting work on developing the event-based drilling machine coordinated with AI for material classification in the context of Industry 4.0. The work is further supported by design analysis and some experimental results. However, the paper needs to undertake the following revisions:

(1) The keywords are better rearranged as 'Drilling; Actuation; Drilling breakthrough; Artificial-Intelligence algorithms; Drilling control'.

(2) Section 4 should be better titled 'Results and discussion'.

(3) Furthermore, section 4 should provide a further clarification and discussion on machining (drilling) trials using the drilling system for some particular engineering materials.

(4) In section 5, the paper should further highlight the novelty of the work presented.

(5) The following very relevant paper in the topic area should be included in References section, particularly against the above comment (4):

Comparative studies on the effect of pilot drillings with application to high speed drilling of carbon fibre reinforced plastic (CFRP) composites, International Journal of Advanced Manufacturing Technology, 89, 2017, 3243-3255.

The paper needs to undertake the following revisions:

(1) The keywords are better rearranged as 'Drilling; Actuation; Drilling breakthrough; Artificial-Intelligence algorithms; Drilling control'.

(2) Section 4 should be better titled 'Results and discussion'.

(3) Furthermore, section 4 should provide a further clarification and discussion on machining (drilling) trials using the drilling system for some particular engineering materials.

(4) In section 5, the paper should further highlight the novelty of the work presented.

(5) The following very relevant paper in the topic area should be included in References section, particularly against the above comments (3) and (4):

Comparative studies on the effect of pilot drillings with application to high speed drilling of carbon fibre reinforced plastic (CFRP) composites, International Journal of Advanced Manufacturing Technology, 89, 2017, 3243-3255.

Author Response

(The authors gave the same response as above.)

Round 2

Reviewer 2 Report

The manuscript " Innovative smart drilling with critical event detection and material classification” presents a cyber-physical drilling machine designed to detect whether it hits or breaks through the workpiece, without the need for additional sensors apart from the position sensor. The results were compared and the conclusions were made.

The authors corrected all items indicated by this reviewer and the manuscript can be accepted and published.

Reviewer 3 Report

The authors provided responses to my comments. I believe that in this version the article can be accepted.